# Invited perspectives: When research meets practice: challenges, opportunities and suggestions from the implementation of the Floods Directive in the largest Italian river basin

By Tommaso Simonelli[1], Laura Zoppi[1], Daniela Molinari[2] & Francesco Ballio[2]

[1] Po River District Authority, Parma, Italy
[2] Department of Civil and Environmental Engineering, Politecnico di Milano, Milano, Italy

*Correspondence to*: Daniela Molinari (daniela.molinari@polimi.it)

Flood damage assessment is a non-consolidated challenging practice for River District Authorities (European Commission, 2016, 2021), which, however, are required to produce flood damage and risk maps to accomplish with the European "Floods Directive". On the other hand, no consolidated standard is available for such evaluations, as flood damage assessment is still an immature topic in the scientific debate (Handmer 2002; Messner and Meyer, 2006; Merz et al. 2010; Gerl et al., 2016; Molinari et al., 2019). In such a context, this manuscript reports challenges, opportunities and perspectives that came into light during the current revision of flood risk maps and flood risk management plans (FRMPs) in the Po River District (Northern Italy, Figure 1), with specific reference to flood damage assessment.

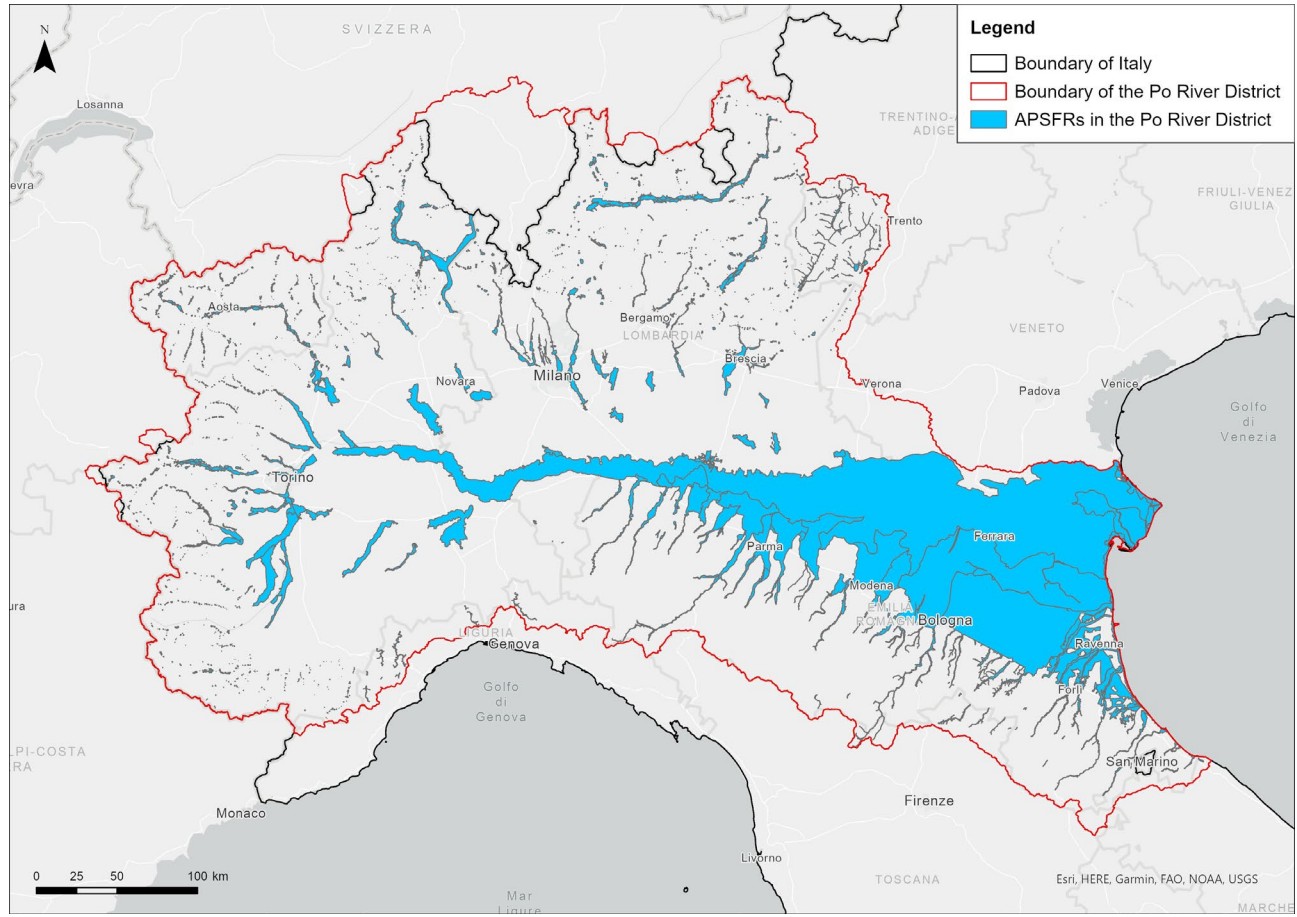

**Figure 1.** Areas of potentially significant flood risk (APFSRs) in the Po River District

The revision process started in January 2020 and is taking place within a partnership lead by the Po River District Authority and including several Italian universities and research centres, coordinated by Politecnico di Milano (i.e., the

MOVIDA project). At present, we are in the conclusion of the project, with first results available on the webpage of the Po River District Authority (AdBPo, 2021). The objective of the project was to identify shared and feasible state of the art solutions for flood damage assessment in the district where, so far, damage and risk were evaluated only in qualitative

terms, mostly according to expert-driven rules of thumbs (Molinari et al., 2016). To this aim the project implemented a case-study, iterative approach (Figure 2). Suitable tools for the assessment of flood damage were first identified among those available in the literature and suitable for the context under investigation, or newly developed (WP_met). Such tools were then tested in 6 pilot areas, characterised by different susceptibility to be damaged in case of flood as well as data availability (WP_6); this step allowed to tune a damage assessment procedure that is flexible and usable in the different

implementation scenarios, which can occur in the district. Once validated, the procedure was implemented in all the areas of potentially significant flood risk (APFSRs) in the district (WP_21A), with the final aim of updating flood damage maps; this was done in collaboration with the main institutions responsible for flood risk management in the district, which were trained in the implementation of the MOVIDA tools (WP_tra), and were supported by the development of a dedicated open Information Systems that allows to implement the procedure in a semi-automatic way (WP_inf). More

details on MOVIDA can be found in the open repository of the project (MOVIDA, 2021)

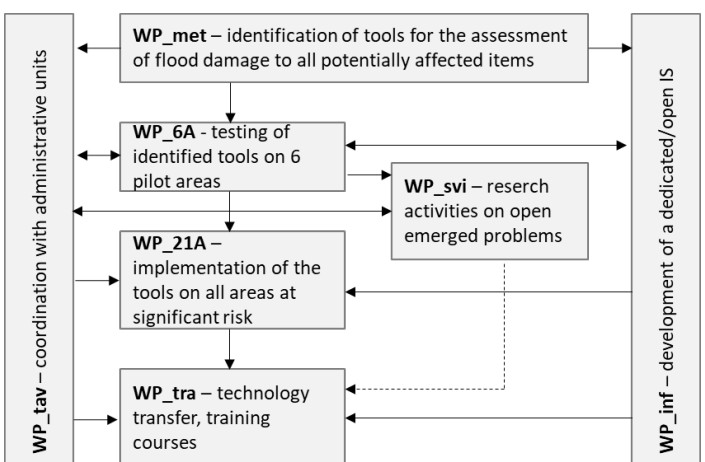

**Figure 2**. Structure of the MOVIDA project

FRMPs ideally require consistent and comprehensive damage assessment for all items which are included in potentially flooded areas, and all kinds of expected impacts, being they related to the direct contact with flood water (i.e., direct damage) or being an indirect consequence of it, like business and services interruption or environmental contamination. Nonetheless, the assessment should lead to a monetary evaluation, to be used as input in Cost Benefit Analyses of alternative mitigation strategies. In practice, this goal is not presently achievable due to the inhomogeneous levels of

development of (and, in some cases the lack of) damage models (for an overview see, e.g., www.fdm.polimi.it, Merz et al. 2010, Pregnolato et al., 2015, Gerl et al., 2016). In particular, in MOVIDA, we were able to identify models for the estimation of direct damage in monetary terms only for residential buildings and a limited number of crops; for economic activities and livestock we could only estimate the exposed economic value (i.e., the maximum potential damage). Scarcity of damage models is also an issue for those items which can be hardly quantified in economic terms (i.e.,

intangible goods) like people, critical infrastructures, cultural heritage and environmental assets. For them, we were able to assess only their amount within the potentially flooded areas, and to classify them according to some vulnerability

features, linked to their susceptibility to be damaged. For the specific case of cultural heritage, given its importance in the Italian context, an ad-hoc procedure was also developed to estimate damage, even though in qualitative terms (i.e., in classes ranges from low to high damage). Indirect damage estimation was instead not feasible, although an attempt has been made to estimate consequences of roads and railways interruption. Indeed, evidence from the past (collected during the project) shows that the weight of direct damage to transport infrastructures is negligible compared to the indirect one.

Paucity and low quality of georeferenced data for the evaluation of characteristics of exposed items further limited the range of damage models that could be implemented for the assessment. In fact, we dealt with scarcity of institutional databases (i.e., data are often stored in commercial repositories), legal impediments in the use of data, fragmentation of information among different databases (even for the same category of elements), and their inadequacy in supplying information required as inputs of the damage models; obsolescence of information was sometimes a problem, with data referring up to ten years ago. A meaningful example of data inadequacy for the Italian context is represented by cultural heritage, for which data are spread among several institutional databases, while a specific asset may be included in more than one database. Moreover, such databases are characterised by different structures, levels of detail, and available information; lack of metadata also hinders their interpretation and comparison.

In general, however, our experience is in line with those of the other European Member States, as can be inferred from Table 1, which summarises how flood damage is currently evaluated within the scope of the Floods Directive. In detail, Table 1 shows how different levels of analysis can currently be achieved for the various items exposed to floods, and how the MOVIDA project allowed the Po River District Authority to implement state of art flood damage modelling tools in the district. In particular, the table highlights the lack of appropriate knowledge and tools for indirect damage assessment.

| ASSETS | | Direct damage | | | Indirect damage |
|---|---|---|---|---|---|
| | | actual damage | potential damage (exposure) | monetary evaluation | |
| people | | | **All MSs, MOVIDA**: number of potentially affected inhabitants on the basis of the flood extent (variability of scales and sources) **SE, BE**: number of people working in the flooded area + number of tourits **MOVIDA:** | no | |
| economic activities | residential buildings | **DK, LT, RO,UK, MOVIDA**: damage to residentail buildings through flood damage models | **EE**: locationing of buildings within the flooded area **Other MSs**: locationing of residential areas within the flood perimeter (from land use maps) | **DK, LT,RO,UK, MOVIDA**: yes **Other MSs**: no | |
| | industrial and commercial activities | **LT**: estimation of expected losses of GDP per person per working day **RO**: damage to activities through flood damage models | **ES**: locationing of activities within the flooded area **Other MSs**: locationing of comm/ind areas within the flood perimeter (from land use maps) **MOVIDA:** monetary values of buildings and contents of activities within the flood perimeter | **LT, RO, MOVIDA**: yes **Other MSs**: no | |
| | agricolture | **DK, RO, MOVIDA**: damage to crops through flood damage models **UK**: damage to agricultural land through one-off cost values | **LT**: damage to crops as lost production **Other MSs**: locationing of agricoltural areas within the flood perimeter (from land use maps) | **DK,RO,UK,LT, MOVIDA**: yes **Other MSs**: no | |
| infrastructure | roads & railways | **DK**: clean-up costs throuh parametric values (clean-up cost/m$^2$) **LT, RO**: repair costs throuh flood damage models | **AT, BE, HR, IT, ES, SI, UK, MOVIDA**: locationing of roads, railway lines, metro lines, metro/train stations within the flooded areas **Other MSs**: locationing of transport infrastructure areas (from land use maps) | **DK,RO,LT**: yes **Other MSs, MOVIDA**: no | expeted impact on lines functionality (qualitative) |
| | strategic buildings/ infrastructure | | **AT, BE, IT, ES, UK, MOVIDA**: locationing of social & health facilities within the flooded area **Other MSs**: locationing of public and infrastructure areas within the flood perimeter (from land use maps) | no | **FI**: consequences of energy disruption on critical services |
| cultural heritage | | **MOVIDA:** qualitative evaluation of damage to cultural heritage assets on the bases of physical vulnerability and importance | **All MSs (but for DK, EE, HU, LU, LV, NL, SK)**: locationing of cultural heritage assets within the flooded perimeter | no | qualitative evaluation of damage to assets on the basis of physical vulnerability and importance |
| protected environmental areas | | | **UK**: Identification of protected areas affected by travelling and dispersion of contaminants from inundated hazardous installations **Other MSs**: locationing of protected areas on the base of the flooded area **MOVIDA:** locationing on the base of flooded perimeter and classification according to ecosystem services provided | no | |
| pollution sources | | | **All MSs, MOVIDA**: locationing of industrial installations within the flooded areas **AT**: classification of installations on the bases of expected impact (qualitative) | no | |

**Table 1:** state of art of flood damage assessment in the European Union within the scope of the Floods Directive at the end of the first implementation cycle, and levels of analysis achievable thanks to the tools developed in the MOVIDA project, for the various assets exposed to flood risk and kinds of damage. Characters in bold refer to ISO country codes, "MSs" stands for Member States (source: European Commission, 2016)

Given the previous premises, the main challenge for damage assessment is the necessity to compare inhomogeneous quantities having different meanings (e.g., damage versus exposed value; direct vs indirect damage) and metrics (e.g., economic loss, physical damage, qualitative damage). Such synthesis is unavoidable if we want to assess the total impact of a flood, as an input for decision making.

What our experience highlights, however, is that the close collaboration between researchers and practitioners allowed to find an equilibrium between scientific rigour and the need of technical improvement. The MOVIDA project led to the identification of feasible solutions to emerged problems and, at the same time, the transferability of scientific knowledge; in this regard, the commitment of several research institutions, working together and sharing knowledge was certainly and added value. For example, thanks to such a collaborative environment we were able to develop tentative models for the estimation of indirect damage to roads and railways; we created an ad-hoc database and procedure for the assessment of damage to cultural heritage; we are presently setting up a model allowing to compare different damage-related data, by the definition of appropriate comparison criteria. The final product of the project is a comprehensive tool allowing for decision-making on flood risk mitigation on the basis of expected risk scenarios, contrary to the present situation when

decisions are taken mostly according to hazard knowledge. At the same time, facing with real problems made researchers aware of limits of available tools, thus proposing new research questions. Starting from such limitations and with the perspective of the next revisions of flood damage maps, we are now developing models for the (quantitative) estimation of flood mortality, indirect damage to people, direct and indirect damage to economic activities, damage to cultural and environmental assets, and damage to infrastructures, also with specific reference to coastal areas (WP_svi in Figure 2).

What clearly emerged from our experience is that flood damage (and, more in general, risk) assessment is not a solely technical problem; social and economic aspects are key elements, calling for a multidisciplinary and a participative approach. Local stakeholders must be especially involved in the final synthesis of the damage evaluations (WP_tav in Figure 2): as previously discussed exposed assets are evaluated through different metrics and suffer from different type of consequences as a result of a flood; the total damage must, therefore, reflect the perception of such values by those who will make use of the assessment in decision-making.

The main conclusion that we can infer from the development of the MOVIDA project is that implementing available and new-coming damage models in real practice is the most appropriate way towards the standardisation of damage assessment tools. Indeed, differently from other disciplines, flood damage models cannot be validated by laboratory tests. Their quality, validity and transferability must be evaluated on the field, and strongly depend on the objective for which the model is implemented, as performing damage assessment for long-term planning purposes may have different requirements than for insurance, or emergency management related reasons. In fact, a model can be very useful for one objective and not for others. To clarify whether a damage model is useful, scientists, practitioners and stakeholders must confront each other, overcoming shared practices. The MOVIDA project represented a good opportunity in this direction. We wish that such a collaborative way of working will be adopted not only in other districts or river basins, but at the European community level. Indeed, in light of the harmonisation required by the European Commission as regards the implementation of the Floods Directive among member states, a comparison between scientists, practitioners and stakeholders at the European level would be suitable, in order to converge on objectives and methods and, in turn, on homogenous requirements of input data at the European level (on which improved datasets can be designed), in a top-down approach; Table 1 shows instead a very fragmented reality as regards flood damage assessment and mapping at the European scale, where the level of analysis achievable for the different exposed assets strongly depends on the availability of national/local tools and required input data. As occurred during the MOVIDA project, the new research challenges and directions will consequently emerge. We claim that such an approach would be beneficial not only for damage assessment related problems but also for challenges linked to other aspects of flood risk management (like climate change), or to the management of other risks. Our wish is the creation of real opportunities to work in this direction, as the definition of a European platform, a COST action or, more ambitiously, an inclusive, big (research) project supported by the European Commission.

**Acknowledgment.** Authors acknowledge with gratitude all researchers involved in the MOVIDA project: C. Armaroli, M. Arosio, R. Giusti, M.L.V. Martina and B. Monteleone from IUSS-Pavia (Italy); C. Arrighi and F. Castelli from University of Florence (Italy); E. Borgogno-Mondino and F. Ghilardi from University of Turin (Italy); F. Carisi, A. Domeneghetti, and N. Petruccelli from University of Bologna (Italy);, P. Ciavola, S. De Biaggi and E. Duo from University of Ferrara (Italy); G. Farina and M. Pilotti from University of Brescia (Italy); A. Gallazzi from Politecnico di Milano (Italy); F. Luino and L. Turconi from CNR-IRPI (Italy); A.R. Scorzini from University of L'Aquila (Italy); M. Hammouti, S. Sterlacchini and M. Zazzeri from CNR-IGAG (Italy).

**Funding.** MOVIDA was funded by the Po River District Authority, and co-funded by the consortium of universities and research centres involved in the project: Politecnico di Milano, University School for Advanced Studies (IUSS) Pavia, University of Florence, University of Turin, University of Bologna, University of Ferrara, University of Brescia, University of l'Aquila, and the National Research Council (CNR)

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
