# Peer review of "Invited perspectives: When research meets practice: challenges, opportunities and suggestions from the implementation of the Floods Directive in the largest Italian river basin"

_Natural Hazards and Earth System Sciences, 2021_

## Author Comment (AC1)

**Manuscript nhess-2021-217: "Invited perspectives: When research meets practice: challenges, opportunities and suggestions from the implementation of the Floods Directive in the largest Italian river basin" – Final response to comments from referee 1**

We would like to thank the referee both for his/her appreciation of our paper and for the work he/she did on our manuscript; we greatly appreciate his/her comments as they may contribute to increase the manuscript robustness and, in general, to improve its quality and readability. In the following, we supply a point by point answer to the comments raised by the referee

**RC1.1** Details of the MOVIDA project are not clear. In order to make it a proper example, it would be relevant to understand:

1) Which institution started the initiative of responding to the requirement of the flood map revision by a project involving all the partners from administrative bodies and research, rather than trying to solve it internally?
2) Was the required funding for MOVIDA just consisting of in-kind contributions, or was it provided by submitting a (research-?) proposal to some funding agency?
3) Is information on the project details available? Please give references.

MOVIDA was started by the Po River District Authority who asked for the support of academia to orient decisions on the adoption of the more suitable tools to assess and map flood damage. The project was partly funded by the same Authority and partly co-funded by the consortium of partners (i.e., universities and research centres).  More information on MOVIDA can be found in the open-repository of the project* and on the website of the Po River District Authority** that, in December 2021, published the results of the implementation of the MOVIDA tools in several areas of potentially significant flood risk (APSFRs) in the District. Such details will be added to the revised version of the paper. Moreover, the description of the different steps of the project, along with a diagram, will be added to answer comments from referee 2.

*                                https://polimi365-my.sharepoint.com/:f:/g/personal/10425403_polimi_it/EvsgjK3x-rJNmnzK5FGg4zYBmtIBXrOVnrzn2kGUamMcVA?e=d3        https://pianoalluvioni.adbpo.it/piano-gestione-rischio-alluvioni-2021KaBG

** https://pianoalluvioni.adbpo.it/piano-gestione-rischio-alluvioni-2021

**RC1.2** (line 26) The website http://www.fdm.polimi.it/ has no relevant content.

We do not agree with the referee. The repository includes many flood damage models available in the literature, classifying them according to the country of development, the damaged element, the kind of damage (i.e., direct vs indirect) considered, etc. This way it supplies a quite exhaustive overview of state of art on flood damage modelling. For this reason, we will leave the reference.

**RC1.3** (line 35 ff) I would also assume that interruptions of electric power and pollution effects (for example, oil spill) should be counted under indirect effects, even if just augmenting other damage. This was observed as a consequence of other flooding events. Have they been relevant in the Po river region?

In fact, evidence collected during the project corroborates literature findings on the importance of electric power disruption and contamination among indirect flood damage (we will quote them as examples in the new version). Still, considering the present paucity of modelling tools for these kinds of damage (that is already discussed in the paper) and the lack of sufficient open data to evaluate the exposure of electric lines, their evaluation was not possible in MOVIDA. Of course, it will be the focus of next research activities.

**RC1.4** (line 40-43) Unavailability of data can also have legal reasons. Is this nothing you came across? The other issue is probably that there are commercial reasons for withholding the data. I think this could be named, rather than just assigning the word "private" for explaining lacking availability.

We are not sure what referee means with "legal". If he/she refers to security reasons that, for example, hamper the publication of sensible data like the location of hazardous installations, this may happen in Italy. We will add this perspective in the revised version of the paper. Likewise, we will accept the suggestion to refer to "commercial" instead of "private data".

**RC1.5** I do not understand what you mean with "complimentary" and existing duplications here.

We mean that the same asset can be present in more than one database. In the new version, we will reframe the sentence to make it clearer.

**RC1.6** (line 49-53) Damage assessment is a precondition for calibration of a damage model (for example, addressing the relation to flood levels). Was the goal just the assessment, or also the calibration and modelling? The latter two are needed for decision making.

We are sorry but we do not understand the comment. In MOVIDA, existing or newly developed damage models are implemented to supply an estimation of expected flood impacts on a certain area at risk, in the support of decision making. The implemented models were chosen among those calibrated and validated (in their developing phase) in contexts that are comparable to the implementation one.

**RC1.7** (line 56) Was it really just the research institutions committing to work together, and not also the administrative institutions? This would actually surprise me.

We are referring here to the commitment to find together (shared) state of the art solutions to emerged problems; of course, knowledge transferability was possible only thanks to the commitment of administrative institutions too.

**RC1.8** (line 89-90) With respect to sustainability, I wonder why you suggest to go for a COST or EU funded project. In the end, it must me administrative bodies of different regions and countries which organize their cooperation. To my knowledge, this is actually a requirement of the European Water Directive, which addresses river basins.

As discussed in the paper, the suggestion goes into the direction of replicating the successful MOVIDA partnership between academia and public institutions in pursuing the objectives of the Floods Directive, calling for the need of upscaling the focus on harmonisation.

---

## Author Comment (AC2)

**Manuscript nhess-2021-217: "Invited perspectives: When research meets practice: challenges, opportunities and suggestions from the implementation of the Floods Directive in the largest Italian river basin" – Final response to comments from referee 2**

We would like to thank the referee both for his/her appreciation of our paper and for the work he/she did on our manuscript; we greatly appreciate his/her comments as they may contribute to increase the manuscript robustness and, in general, to improve its quality and readability. In the following, we supply a point by point answer to the comments raised by the referee

**RC2.1** a flow diagram representing the main components of the MOVIDA project is highly recommended; otherwise, the readers will not fully understand how articulated is the project

A description of the main steps of the project and an explicative diagram will be added to the revised version of the manuscript

**RC2.2** a section (just half a page, no more since this is a short perspective) on the advance of MOVIDA with respect to other similar projects (not only in Italy) is recommended; an improvement of literature is necessary

A critical comparison of damage assessment performed in MOVIDA and tools/methods presently implemented in the various European Member States to assess and map flood damage within the scope of the Floods Directive will be added to the new version of the manuscript. Discussion will be supported by a summary table. On the contrary, we do not think this is the place to discuss literature on flood damage modelling.

**RC2.3** a mention (in the manuscript, not in the acknowledgement) of all the institutions involved in MOVIDA is recommended, maybe adding these in the above-mentioned flow diagram

We think this option would take the reader away from the main core of the paper. However, we will add a funding section where we list all involved institutions, also to answer comment from referee 1

**RC2.4** an additional figure (location map) on the study area (the Po River basin), with 2-3 popups representative location subject to floods (occurred events) and/or at risk of floods, could also help in improving the value and readership.

A map of the District with the identification of the potentially flooded areas will be added in the revised version of the paper

---

## Author Response (AR2)

**Manuscript nhess-2021-217: "Invited perspectives: When research meets practice: challenges, opportunities and suggestions from the implementation of the Floods Directive in the largest Italian river basin" – Final response to comments from referee 1**

We would like to thank the referee and the editor for this second round of review. In the following, we supply a point by point answer to the comments further raised by the referee.

**RC1.1** A main point here is the overarching relevance of the project as a demonstration example in order to provide a "perspective" for research activities on Natural Hazards in a more general sense. While it is mentioned in the text that the initial impetus came from formal requirement of a revision of flood risk maps and flood risk management plans, a positive effect on the availability of impact data for the setup and calibration of damage models arising from the involvement of the respective authorities is still not clearly spelled out.

As already mentioned in the manuscript, the project provided many perspectives for research activities, highlighting gaps and needs on which research is currently under development, e.g., regarding the (quantitative) estimation of flood mortality, indirect damage to people, direct and indirect damage to economic activities, damage to cultural and environmental assets, and damage to infrastructures, also with specific reference to coastal areas (lines 93-96). With respect to the production or gathering of "new" impact data, by considering also comment RC1.6 made by the reviewer in the first round, we suspect he/she did not really understand the objective of MOVIDA, that is not the ex-post damage assessment. In MOVIDA, existing or newly developed damage models are implemented to supply an estimation of expected flood impacts on a certain area at risk, in the support of decision making. The implemented models were chosen among those calibrated and validated (in their developing phase) in contexts that are comparable to the implementation one.

**RC1.2** The website provides just a blank page without content. I tried this out using different web browsers. Please check!

We are really sorry, but we did not realise the webpage was out of order. We are fixing the problem, but it still requires some days. If the manuscript will be accepted and the problem will not be fixed yet, we commit to remove the reference during the proofing correction stage. We also added further references to complement the existing one (line 45-46)

**RC1.3** I am referring to your last sentence: "Our wish is the creation of real opportunities to work in this direction, as the definition of a European platform, a COST action or,more ambitiously, an inclusive, big (research) project supported by the European Commission.!". I do not understand this statement in the context of your manuscript and of your response. While you suggest replicating the partnership between academia and public institutions, it is not clear what the role of COST actions or big research projects should be. Rather, such initiatives could be started on a regional basis, for example motivated by demands from the Floods Directive.

As the referee highlights, we suggest and wish to replicate the partnership between academia and institutions at the European Level, just to find common solutions to Floods Directive (and more in general Flood Risk Management) requirements. The development of a European platform, a COST action, or an EU project is, in our opinion, a concrete way to support the creation of such partnership, both from an economic and a logistic perspective. As said, it is just our opinion. We leave the editor to decide about the appropriateness of the sentence for the scope of the special issue. If it is not appropriate, we will simply delate the sentence.